# Generative 3D Part Assembly via Dynamic Graph Learning

**Jialei Huang**[*]
Peking University
1600012454@pku.edu.cn

**Guanqi Zhan**[*]
Peking University
guanqi_zhan@pku.edu.cn

**Qingnan Fan**
Stanford University
fqnchina@gmail.com

**Kaichun Mo**
Stanford University
kaichun@cs.stanford.edu

**Lin Shao**
Stanford University
lins2@stanford.edu

**Baoquan Chen**
CFCS, CS Dept., Peking University
AIIT, Peking University
baoquan.chen@gmail.com

**Leonidas Guibas**
Stanford University
guibas@cs.stanford.edu

**Hao Dong**[†]
CFCS, CS Dept., Peking University
AIIT, Peking University
Peng Cheng Laboratory
hao.dong@pku.edu.cn

## Abstract

Autonomous part assembly is a challenging yet crucial task in 3D computer vision and robotics. Analogous to buying an IKEA furniture, given a set of 3D parts that can assemble a single shape, an intelligent agent needs to perceive the 3D part geometry, reason to propose pose estimations for the input parts, and finally call robotic planning and control routines for actuation. In this paper, we focus on the pose estimation subproblem from the vision side involving geometric and relational reasoning over the input part geometry. Essentially, the task of generative 3D part assembly is to predict a 6-DoF part pose, including a rigid rotation and translation, for each input part that assembles a single 3D shape as the final output. To tackle this problem, we propose an assembly-oriented dynamic graph learning framework that leverages an iterative graph neural network as a backbone. It explicitly conducts sequential part assembly refinements in a coarse-to-fine manner, exploits a pair of part relation reasoning module and part aggregation module for dynamically adjusting both part features and their relations in the part graph. We conduct extensive experiments and quantitative comparisons to three strong baseline methods, demonstrating the effectiveness of the proposed approach.

## 1 Introduction

It is a complicated and laborious task, even for humans, to assemble an IKEA furniture from its parts. Without referring to any procedural or external guidance, *e.g.* reading the instruction manual, or watching a step-by-step video demonstration, the task of 3D part assembly involves exploring an extremely large solution spaces and reasoning over the input part geometry for candidate assembly proposals. To assemble a physically stable furniture, a rich set of part relations and constraints need to be satisfied for a successful assembly.

---

[*]Equal contribution. Author ordering determined by coin flip.
[†]Corresponding author.

There are some literature in the computer vision and graphics fields that study part-based 3D shape modeling and synthesis. For example, [3, 29, 30] employ a third-party repository of 3D meshes for part retrieval to assemble a complete shape. Benefiting from recent large-scale part-level datasets [24, 38] and the advent of deep learning techniques, some recent works [14, 27, 34] leverage deep neural networks to sequentially generate part geometry and posing transform for shape composition. Though similar to our task, none of these works addresses the exactly same setting to ours. They either allow free-form part generation for the part geometry, or assume certain part priors, such as an available large part pool, a known part count and semantics for the entire category of shapes (*e.g.*, for chairs, four semantic parts: back, seat, leg and arm). In our setting, we assume no semantic knowledge upon the input parts and assemble 3D shapes conditioned on a given set of fine-grained part geometry with variable number of parts for different shape instances.

In this paper, we propose to use a dynamic graph learning framework that predicts a 6-DoF part pose, including a rigid rotation and translation, for each input part point cloud via forming a dynamically varying part graph and iteratively reasoning over the part poses and their relations. We employ an iterative graph neural network to gradually refine the part pose estimations in a coarse-to-fine manner. At the core of our method, we propose the dynamic part relation reasoning module and the dynamic part aggregation module that jointly learns to dynamically evolve part node and edge features within the part graph.

Lack of the real-world data for 3D part assembly, we train and evaluate the proposed approach on the synthetic PartNet dataset, which provides a large-scale benchmark with ground-truth part assembly for ShapeNet models at the fine-grained part granularity. Although there is no previous work studying the exactly same problem setting as ours, we formulate three strong baselines inspired by previously published works on similar task domains and demonstrate that our method outperforms baseline methods by significant margins.

Diagnostic analysis further indicates that in the iterative part assembly procedure, a set of central parts (*e.g.* chair back, chair seat) learns much faster than the other peripheral parts (*e.g.* chair legs, chair arms), which quickly sketches out the shape backbone in the first several iterations. Then, the peripheral parts gradually adjust their poses to match with the central part poses via the graph message-passing mechanism. Such dynamic behaviors are automatically emerged without direct supervision and thus demonstrate the effectiveness for our dynamic graph learning framework.

## 2  Related Work

**Assembly-based 3D modeling.** Part assembly is an important task in many fields. Recent works in the robotic community [19, 39, 28, 20] emphasize the planning, in-hand manipulation and robot grasping using a partial RGB-D observation in an active learning manner, while our work shares more similarity with the work in the vision and graphics background, which focuses on the problem of pose or joint estimation for part assembly. On this side, [7] is the pioneering work to construct 3D geometric surface models by assembling parts of interest in a repository of 3D meshes. The follow-ups [2, 11, 10] learn a probabilistic graphical model that encodes semantic and geometric relationships among shape components to explore the part-based shape modeling. [3, 29, 30] model the 3D shape conditioned on the single-view scan input, rough models created by artists or a partial shape via an assembly manner.

However, most of these previous works rely on a third-part shape repository to query a part for the assembly. Inspired by the recent generative deep learning techniques and benefited from the large-scale annotated object part datasets [24, 38], some recent works [14, 27, 34] generate the parts and then predict the per-part transformation to compose the shape. [5] introduces a Decomposer-Composer network for a novel factorized shape latent space. These existing data-driven approaches mostly focus on creating a novel shape from the accumulated shape prior, and base the estimated transformation parameters on the 6-DoF part pose of translation and scale. They assume object parts are well rotated to stand in the object canonical space. In this work, we focus on a more practical problem setting, similar to assembling parts into a furniture in IKEA, where all the parts are provided and laid out on the ground in the part canonical space. Our goal of part assembly is to estimate the part-wise 6-DoF pose of rotation and translation to compose the parts into a complete shape (furniture). A recent work [17] has a similar setting but requires an input image as guidance.

**Structure-aware generative networks.** Deep generative models, such as generative adversarial networks (GAN) [9] and variational autoencoders (VAE) [12], have been explored recently for shape generation tasks. [15, 21] propose hierarchical generative networks to encode structured models, represented as abstracted bounding box. The follow-up work [22] extends the learned structural variations into conditional shape editing. [8] introduces a two-level VAE to jointly learns the global shape structure and fine part geometries. [35] proposes a two-branch generative network to exchange information between structure and geometry for 3D shape modeling. [31] presents a global-to-local adversarial network to construct the overall structure of the shape, followed by a conditional autoencoder for part refinement. Recently, [23] employs a conditional GAN to generate a point cloud from an input rough shape structure. Most of the aforementioned works couple the shape structure and geometry into the joint learning for diverse and perceptually plausible 3D modeling. However, we focus on a more challenging problem that aims at generating shapes with only structural variations conditioned on the fixed detailed part geometry.

# 3  Assembly-Oriented Dynamic Graph Learning

Given a set of 3D part point clouds $\mathcal{P} = \{p_i\}_{i=1}^N$ as inputs, where $N$ denotes the number of parts which may vary for different shapes, the goal of our task is to predict a 6-DoF part pose $q_i \in SE(3)$ for each input part $p_i$ and form a final part assembly for a complete 3D shape $S = \cup_i q_i(p_i)$, where $q_i(p_i)$ denotes the transformed part point cloud $p_i$ according to $q_i$.

To tackle this problem, we propose an assembly-oriented dynamic graph learning framework that leverages an iterative graph neural network (GNN) as a backbone, which explicitly conducts sequential part assembly refinements in a coarse-to-fine manner, and exploits a pair of part relation reasoning module and part aggregation module for iteratively adjusting part features and their relations in the part graph. Figure 1 illustrates our proposed pipeline. Below, we first introduce the iterative GNN backbone and then discuss the dynamic part relation reasoning module and part aggregation module in detail.

## 3.1  Iterative Graph Neural Network Backbone

We represent the dynamic part graph at every time step $t$ as a self-looped directed graph $\mathcal{G}^{(t)} = (\mathcal{V}^{(t)}, \mathcal{E}^{(t)})$, where $\mathcal{V}^{(t)} = \{v_i^{(t)}\}_{i=1}^N$ is the set of nodes and $\mathcal{E}^{(t)} = \{e_{ij}^{(t)}\}$ is the set of edges in $\mathcal{G}^{(t)}$. We treat each part as a node in the graph and initialize its attribute $v_i^{(0)} \in \mathbb{R}^{256}$ via encoding the part geometry $p_i \in \mathbb{R}^{1000 \times 3}$ as $v_i^{(0)} = f_{init}(p_i)$, where $f_{init}$ is a parametric function implemented as a vanilla PointNet [25] that extracts a global permutation-invariant feature summarizing the input part point cloud $p_i$. We use a shared PointNet $f_{init}$ to process all the parts.

We use a fully connected graph, drawing an edge among all pairs of parts, and perform the iterative graph message-passing operations via alternating between updating the edge and node features. To be specific, the edge attribute $e_{ij}^{(t)} \in \mathbb{R}^{256}$ emitting from node $v_j^{(t)}$ to $v_i^{(t)}$ at time step $t$ is calculated as a neural message

$$e_{ij}^{(t)} = f_{edge}(v_i^{(t)}, v_j^{(t)}), \tag{1}$$

which is then leveraged to update the node attribute $v_i^{(t+1)}$ at the next time step $t+1$ by aggregating messages from all the other nodes

$$v_i^{(t+1)} = f_{node}(v_i^{(t)}, \frac{1}{N} \sum_{j=1}^N e_{ij}^{(t)}), \tag{2}$$

that takes both the previous node attribute and the averaged message among neighbors as inputs. The part pose $q_i^{(t+1)} \in \mathbb{R}^7$, including a 3-DoF rotation represented as a unit 4-dimensional Quaternion vector and a 3-dimensional translation vector denoting the part center offset, is then regressed by decoding the updated node attribute via

$$q_i^{(t+1)} = f_{pose}(v_i^{(0)}, v_i^{(t+1)}, q_i^{(t)}). \tag{3}$$

Besides the node feature at the current time step $v_i^{(t+1)}$, $f_{pose}$ also takes as input the initial node attribute $v_i^{(0)}$ to capture the raw part geometry information, and the estimated pose in the last time

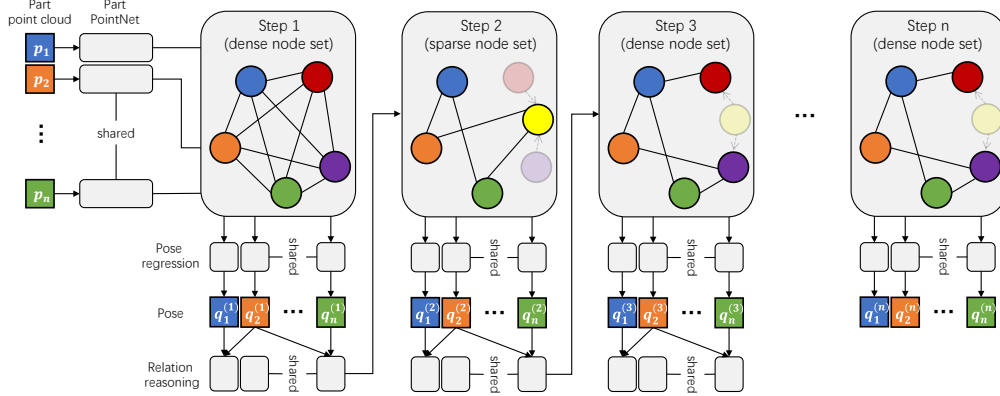

Figure 1: The proposed dynamic graph learning framework. The iterative graph neural network backbone takes a set of part point clouds as inputs and conducts 5 iterations of graph message-passing for coarse-to-fine part assembly refinements. The graph dynamics is encoded into two folds, (a) reasoning the part relation (graph structure) from the part pose estimation, which in turn also evolves from the updated part relations, and (b) alternatively updating the node set by aggregating all the geometrically-equivalent parts (the red and purple nodes), *e.g.* two chair arms, into a single node (the yellow node) to perform graph learning on a sparse node set for even time steps, and unpooling these nodes to the dense node set for odd time steps. Note the semi-transparent nodes and edges are not included in graph learning of certain time steps.

step $q_i^{(t)}$ for more coherent pose evolution. Note that $q_i^{(0)}$ is not defined and hence not inputted to $f_{pose}$ at the first time step.

In our implement, $f_{edge}$, $f_{node}$ and $f_{pose}$ are all parameterized as Multi-Layer Perceptrons (MLP) that are shared across all the edges or nodes for each time step. Note that we use different network weights for different iterations, since the node and edge features evolve over time and may contain information at different scales. Our iterative graph neural network runs for 5 iterations and learns to sequentially refine part assembly in a coarse-to-fine manner.

## 3.2 Dynamic Relation Reasoning Module

The relationship between entities is known to be important for a better understanding of visual data. There are various relation graphs defined in the literature. [36, 18, 37, 4] learn the scene graph from the labeled object relationship in a large-scale image dataset namely Visual Genome [13], in favor of the 2D object detection task. [16, 41, 32, 26] calculate the statistical relationships between objects via some geometrical heuristics for the 3D scene generation problem. In terms of the shape understanding, [21, 8] define the relation as the adjacency or symmetry between every two parts in the full shape.

In our work, we learn dynamically evolving part relationships for the task of part assembly. Different from many previous dynamic graph learning algorithms [33, 40] that only evolve the node and edge features implicitly, we propose to update the relation graph based on the estimated part assembly at each time step explicitly. This is special for our part assembly task and we incorporate the assembly-flavor in our model design. At each time step, we predict the part pose for each part and the obtained temporary part assembly enables explicit reasoning about how to refine part poses in the next step considering the current part disconnections and the overall shape geometry.

To achieve this goal, besides the maintained edge attributes, we also learn to reason a directed edge-wise weight scalar $r_{ij}^{(t)}$ to indicate the significance of the relation from node $v_j^{(t)}$ to $v_i^{(t)}$. Then, we update the node attribute at time step $t+1$ via multiplying the weight scalar and edge attribute

$$v_i^{(t+1)} = f_{node}\left(v_i^{(t)}, \frac{\sum_j e_{ij}^{(t)} r_{ij}^{(t)}}{\sum_j r_{ij}^{(t)}}\right). \tag{4}$$

There are various options to implement $r_{ij}^{(t)}$. For example, one can employ the part geometry transformed with the estimated poses to regress the relation, or incorporate the holistic assembled shape feature. In our implementation, however, we find that directly exploiting the pose features to learn the relation $r_{ij}^{(t)}$ is already satisfactory. This is mainly caused by the fact that the parts of different semantics may share similar geometries but usually have different poses. For instance, the leg and leg stretcher are geometrically similar but placed and oriented very differently. Therefore, we adopt the simple solution by reasoning the relation $r_{ij}^{(t)}$ only from the estimated poses

$$r_{ij}^{(t)} = f_{relation}\left(f_{feat}\left(q_i^{(t-1)}\right), f_{feat}\left(q_j^{(t-1)}\right)\right), \tag{5}$$

where both $f_{feat}$ and $f_{relation}$ are parameterized as MLPs. $f_{feat}$ is used to extract the independent pose feature from each part pose prediction. Note that we set $r_{ij}^{(1)} = 1$ in the beginning.

### 3.3 Dynamic Part Aggregation Module

The input parts may come in many geometrically-equivalent groups, *e.g.*, four chair legs, two chair arms, where the parts in each group share the same part geometry and hence can be detected via some simple heuristics. We observe that geometrically-equivalent parts are highly correlated and thus very likely to share common knowledge regarding part poses and relationships. For example, four long sticks may all serve as legs that stand upright on the ground, or two leg stretchers that are parallel to each other. Thus, we propose a dynamic part aggregation module that allows more direct information exchanges among geometrically-equivalent parts.

To achieve this goal, we explicitly create two sets of nodes at different assembly levels: a dense node set including all the part nodes, and a sparse node set created by aggregating all the geometrically-equivalent nodes into a single node. Then, we perform the graph learning via alternatively updating the relation graph between the dense and sparse node sets. In this manner, we allow dynamic communications among geometrically-equivalent parts for synchronizing shared information while learning to diverge to different part poses.

In our implementation, we denote $\mathcal{G}^{(t)}$ as the dense node graph at time step $t$. To create a sparse node graph $\mathcal{G}^{(t+1)}$ at time step $t + 1$, we firstly aggregate the node attributes among the geometrically-equivalent parts $\mathcal{V}_g$ via max-pooling into a single node $v_j^{(t)}$ as

$$v_j^{(t)} = \underset{k \in \mathcal{V}_g}{\text{pooling}}\left(v_k^{(t)}\right) \tag{6}$$

Then, we aggregate the relation weights from geometrically-equivalent parts to any other node $v_i^{(t)}$

$$r_{ij}^{(t)} = f_{relation}\left(f_{feat}\left(q_i^{(t-1)}\right), \underset{k \in \mathcal{V}_g}{\text{pooling}}\left(f_{feat}\left(q_k^{(t-1)}\right)\right)\right). \tag{7}$$

The inverse relation emitted from the node $v_i^{(t)}$ to the aggregated node $v_j^{(t)}$ is computed similarly as

$$r_{ji}^{(t)} = f_{relation}\left(\underset{k \in \mathcal{V}_g}{\text{pooling}}\left(f_{feat}\left(q_k^{(t-1)}\right)\right), f_{feat}\left(q_i^{(t-1)}\right)\right). \tag{8}$$

All these equations are conducted once we finish the update of dense node graph $\mathcal{G}^{(t)}$, then we are able to operate on the sparse node graph $\mathcal{G}^{(t+1)}$ following Eq. (4) and (5). To enrich the sparse node set back to dense node set, we simply unpool the node features to a corresponding set of geometrically-equivalent parts. We alternatively conduct dynamic graph learning over the dense and sparse node sets at odd and even iterations separately.

### 3.4 Training and Losses.

Given an input set of part point clouds, there may be multiple solutions for the shape assembly. For example, one can move a leg stretcher up and down as long as it is connected to two parallel legs. The chair back can also be possibly laid down to form a deck chair. To address the multi-modal

predictions, we employ the Min-of-N (MoN) loss [6] to balance between the assembly quality and diversity. Let $f$ denote our whole framework, which takes in the part point cloud set $\mathcal{P}$ and a random noise $z_j$ sampled from unit Gaussian distribution $\mathcal{N}(0, 1)$. Let $\mathcal{L}$ be any loss function supervising the network outputs $f(\mathcal{P}, z_j)$ and $f^*(\mathcal{P})$ be one provided ground truth sample in the dataset, then the MoN loss is defined as

$$\min_{z_j \sim \mathcal{N}(0,1)} \mathcal{L}\left(f(\mathcal{P}, z_j), f^*(\mathcal{P})\right). \tag{9}$$

The MoN loss encourages at least one of the predictions to be close to the ground truth data, which is more tolerant to the dataset of limited diversity and hence more suitable for our problem. In practice, we sample 5 particles of $z_j$ to approximate Eq. (9).

The $\mathcal{L}$ is implemented as a weighted combination of both local part and global shape losses, detailed as below. Each part pose $q_i$ can be decomposed into rotation $r_i$ and translation $t_i$. We supervise the translation via an $\mathcal{L}_2$ loss,

$$\mathcal{L}_t = \sum_{i=1}^{N} ||t_i - t_i^*||_2^2 \tag{10}$$

The rotation is supervised via Chamfer distance on the rotated part point cloud

$$\mathcal{L}_r = \sum_{i=1}^{N} \left( \sum_{x \in q_i(p_i)} \min_{y \in q_i^*(p_i)} ||x - y||_2^2 + \sum_{x \in q_i^*(p_i)} \min_{y \in q_i(p_i)} ||x - y||_2^2 \right). \tag{11}$$

In order to achieve good assembly quality holistically, we also supervise the full shape assembly $S$ using Chamfer distance (CD),

$$\mathcal{L}_s = \sum_{x \in S} \min_{y \in S^*} ||x - y||_2^2 + \sum_{x \in S^*} \min_{y \in S} ||x - y||_2^2. \tag{12}$$

In all equations above, the asterisk symbols denote the corresponding ground-truth values.

## 4 Experiments and Analysis

We conduct extensive experiments demonstrating the effectiveness of the proposed method and show quantitative and qualitative comparisons to three baseline methods. We also provide diagnostic analysis over the learned part relation dynamics, which clearly illustrates the iterative coarse-to-fine refinement procedure.

### 4.1 Dataset

We leverage the recent PartNet [24], a large-scale shape dataset with fine-grained and hierarchical part segmentations, for both training and evaluation. We use the three largest categories, chairs, tables and lamps, and adopt its default train/validation/test splits in the dataset. In total, there are 6,323 chairs, 8,218 tables and 2,207 lamps. We deal with the most fine-grained level of PartNet segmentation. We use Furthest Point Sampling (FPS) to sample 1,000 points for each part point cloud. All parts are zero-centered and provided in the canonical part space computed using PCA.

### 4.2 Baseline Approaches

Since our task is novel, there is no direct baseline method to compare. However, we try to compare to three baseline methods inspired by previous works sharing similar spirits of part-based shape modeling or synthesis. The baseline methods are trained using the same losses and the same termination strategy as our method. We stop training when they achieve the best scores on the validation set.

**B-Complement:** ComplementMe [30] studies the task of synthesizing 3D shapes from a big repository of parts and mostly focus on retrieving part candidates from the part database. We modify the setting to our case by limiting the part repository to the input part set and sequentially predicting a part pose for each part.

**B-LSTM:** Instead of leveraging a graph structure to encode and decode part information jointly, we use a bidirectional LSTM module similar to PQ-Net [34] to sequentially estimate the part pose. Note

that the original PQ-Net studies the task of part-aware shape generative modeling, which is a quite different task from ours.

**B-Global:** Without using the iterative GNN, we directly use the per-part feature, augmented with the global shape descriptor, to regress the part pose in one shot. Though dealing with different tasks, this baseline method borrows similar network design with CompoNet [27] and PAGENet [14].

## 4.3 Evaluation Metrics

We use the Minimum Matching Distance (MMD) [1] to evaluate the fidelity of the assembled shape. Conditioned on the same input set of parts, we generate multiple shapes sampled from different Gaussian noises, and measure the minimum distance between the ground truth and the assembled shapes. We adopt three distance metrics, *part accuracy*, *shape chamfer distance* following [17] and *connectivity accuracy* proposed by us. The part accuracy is defined as,

$$\frac{1}{N}\sum_{i=1}^{N}\mathbb{1}\left(\left(\sum_{x\in q_i(p_i)}\min_{y\in q_i^*(p_i)}||x-y||_2^2 + \sum_{x\in q_i^*(p_i)}\min_{y\in q_i(p_i)}||x-y||_2^2\right) < \tau_p\right) \quad (13)$$

where we pick $\tau_p = 0.01$. Intuitively, it indicates the percentage of parts that match the ground truth parts to a certain CD threshold. Shape chamfer distance is calculated the same as Eq. 12.

**Connectivity Accuracy.** The part accuracy measures the assembly performance by considering each part separately. In this work, we propose the *connectivity accuracy* to further evaluate how well the parts are connected in the assembled shape. For each connected part pair $<p_i^*, p_j^*>$ in the object space, we firstly select one point in part $p_i^*$ that is closest to part $p_j^*$ as $p_i^*$'s contact point $c_{ij}^*$ with respect to $p_j^*$, then select the point in $p_j^*$ that is closest to $c_{ij}^*$ as the corresponding $p_j^*$'s contact point $c_{ji}^*$. Given the predefined contact point pair $\{c_{ij}^*, c_{ji}^*\}$ located in the object space, we transform each point into its corresponding canonical part space as $\{c_{ij}, c_{ji}\}$. Then we calculate the connectivity accuracy of an assembled shape as

$$\frac{1}{N_{cp}}\sum_{\{c_{ij}, c_{ji}\}\in C}\mathbb{1}\left(||q_i(c_{ij}) - q_j(c_{ji})||_2^2 < \tau_c\right), \quad (14)$$

where $C$ denotes the set of contact point pairs $\{c_{ij}, c_{ji}\}$ and $\tau_c = 0.01$. It evaluates the percentage of correctly connected parts.

## 4.4 Results and Comparisons

We present the quantitative comparisons with the baselines in Table 1. Our algorithm outperforms all these approaches by a significant margin for most columns, especially on the part and connectivity accuracy metrics. According to the visual results in Figure 2 (left), we also observe the best assembly results are achieved by our algorithm, while the baseline methods usually fail in producing well-structured shapes. We generate multiple possible assembly outputs with different Gaussian random noise, and measure the minimum distance from the assembly predictions to the ground truth, which allows the models to generate shapes that are different from ground truth while similar to real-world objects, which are demonstrated in Figure 2 (right). We see that some bar-shape parts are assembled into different positions to form objects of different structures.

| | Shape Chamfer Distance ↓ | | | Part Accuracy ↑ | | | Connectivity Accuracy ↑ | | |
|---|---|---|---|---|---|---|---|---|---|
| | Chair | Table | Lamp | Chair | Table | Lamp | Chair | Table | Lamp |
| B-Global | 0.0146 | 0.0112 | 0.0079 | 15.7 | 15.37 | 22.61 | 9.90 | 33.84 | 18.6 |
| B-LSTM | 0.0131 | 0.0125 | **0.0077** | 21.77 | 28.64 | 20.78 | 6.80 | 22.56 | 14.05 |
| B-Complement | 0.0241 | 0.0298 | 0.0150 | 8.78 | 2.32 | 12.67 | 9.19 | 15.57 | 26.56 |
| **Ours** | **0.0091** | **0.0050** | 0.0093 | **39.00** | **49.51** | **33.33** | **23.87** | **39.96** | **41.70** |

Table 1: Quantitative Comparison between our approach and the baseline methods.

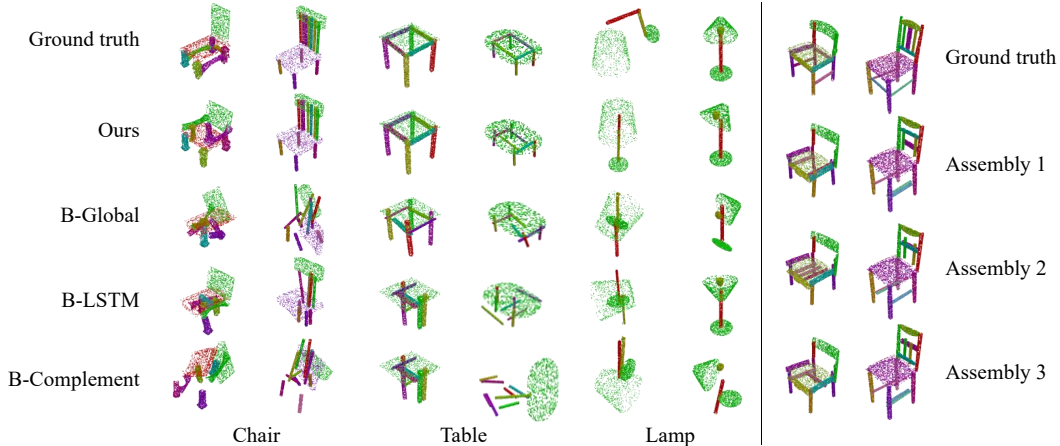

Figure 2: Qualitative Results. Left: visual comparisons between our algorithm and the baseline methods; Right: multiple plausible assembly results generated by our network.

|  | Shape Chamfer Distance ↓ | Part Accuracy ↑ | Connectivity Accuracy ↑ |
|---|---|---|---|
| Our backbone w/o graph learning | 0.0086 | 26.05 | 28.07 |
| Our backbone | 0.0055 | 42.09 | 35.87 |
| Our backbone + relation reasoning | 0.0052 | 46.85 | 38.60 |
| Our backbone + part aggregation | 0.0051 | 48.01 | 38.13 |
| Our full algorithm | **0.0050** | **49.51** | **39.96** |

Table 2: Experiments demonstrate that all the three key components are necessary.

We also try to remove the three key components from our method: the iterative GNN, the dynamic part relation reasoning module and the dynamic part aggregation module. The results on the PartNet Table category are shown in Table 2 and we see that our full model achieves the best performance compared to the ablated versions. We firstly justify the effectiveness of our backbone by replacing the graph learning module with a multi-layer perception that estimates the part-wise pose from the concatenated separate and overall part features. We further incorporate the dynamic graph module into our backbone for evaluation. We observe that our proposed backbone and dynamic graph both contribute to the final performance significantly.

To further investigate the different design choices of our algorithm, we conduct more ablation studies on the Table category regarding the iteration number in GNN (3/5/7, PA: 45.51/49.51/44.6), the number of input points for each part (500/1000, PA: 48.78/49.51), average/max pooling in the part aggregation module (average/max, PA: 48.92/49.51), and different loss designs (w/o $\mathcal{L}_t/\mathcal{L}_r/\mathcal{L}_s$, PA: 24.48/46.94/48.23). PA is short for part accuracy. We find empirically that having too many iterations, makes the network harder to train and difficult to convergence, and hence finally set the iteration number to 5 in our paper. Taking more per-part input points, and using max pooling instead of average pooling both improve the performance. The three loss terms all contribute to the final results.

## 4.5 Dynamic Graph Analysis

Figure 3 summarizes the learned directional relation weights at each time step by averaging $r_{ij}^{(t)}$ over all PartNet chairs. We pick the four common types of parts: back, seat, leg and arm. We see clearly similar statistical patterns for the even iterations and for the odd ones. At even iterations, the relation graph is updated from the dense node set. It focuses more on passing messages from the central parts (*i.e.* seat, back) to the peripheral parts (*i.e.* leg, arm) and overlooks the relation between legs. While at odd iterations, the relation graph is updated from the sparse node set, where the geometrically-equivalent parts like legs are aggregated to a single node. In this case, from the relation graph we can see that all the parts are influenced relatively more by the peripheral parts. Note legs are usually

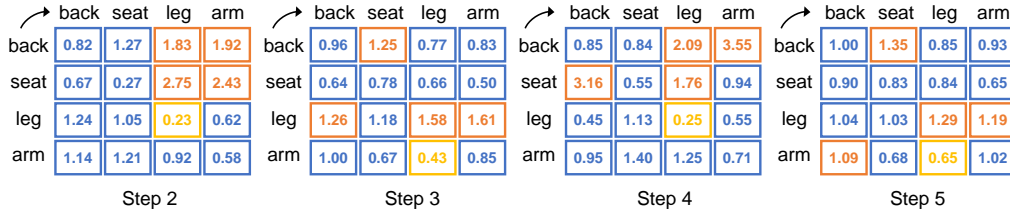

Figure 3: Dynamically evolving part relation weights $r_{ij}^{(t)}$ among four common chair part types. The orange cells highlight the four directed edges with the maximal learned relation weight in the matrix, while the yellow cells indicate the minimal ones. The vertical axis denotes the emitting parts, and the horizontal axis denotes the receiving parts.

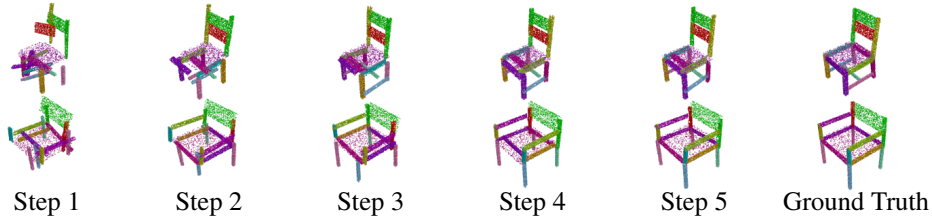

Figure 4: The time-varying part assembly results.

symmetric, so it is very likely that they share the same/similar part orientations, which explains their strong correlations here. On the average of all the steps, the central parts have bigger emitting relation weights (1.22) than the peripheral parts (0.94), indicating that the central parts guide the assembly process more.

We further illustrate the changes of part accuracy and its associated improvement at each time step in Figure 5. We find that the central parts are consistently predicted more accurately than the peripheral parts. Interestingly, the improvement of peripheral parts is relatively higher than central parts at even iterations, which is consistent to the observation of higher directed relations received to minor parts at odd iterations. It demonstrates the fact that central parts guide the pose predictions for the peripheral parts.

Figure 4 visualizes the time-varying part assembly results, showing that the poses for the central parts are firstly determined and then the peripheral parts gradually adjust their poses to match the central parts. The results finally converge to stable part assembly predictions.

# 5 Conclusion

In this work, we propose a novel dynamic graph learning algorithm for the part assembly problem. We develop an iterative graph neural network backbone that learns to dynamically evolve node and edge features within the part graph, augmented with the dynamic relation reasoning module and the dynamic part aggregation module. Through thorough experiments and analysis, we demonstrate that the proposed method achieves state-of-the-art performance over the three baseline methods by learning an effective assembly-oriented relation graph. Future works may investigate learning better part assembly generative models considering the part joint information and higher-order part relations.

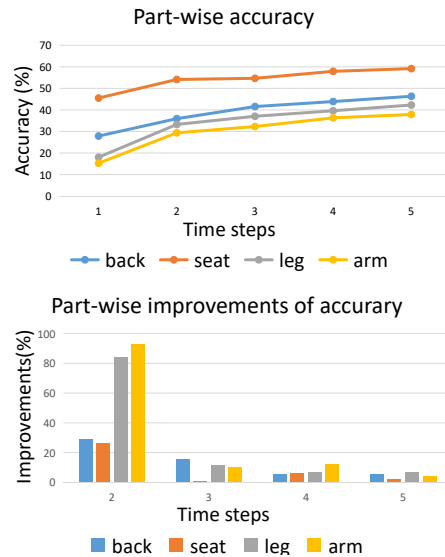

Figure 5: The part accuracy and its relative improvement summarized at each time step.

## Broader Impact

Automatic part assembly is an important task in robotics. Nowadays, tens of thousands of production line workers are working on product assembly manually. Learning how to assemble a set of parts to a complete shape is still a challenging problem. This study can enable robots to free human on part assembly task. We believe this research benefits both the economy and society.

However, even with learning algorithms, human trust in machines is still a problem. The algorithms can only be used for assistance in practice and cannot fully replace humans. In the assembly task, such machine learning algorithm/system needs to cooperate with human to achieve more effective processes.

## Acknowledgement

This work was supported by the start-up research funds from Peking University (7100602564) and the Center on Frontiers of Computing Studies (7100602567), China National Key R&D Program (2016YFC0700100, 2019YFF0302902), a Vannevar Bush Faculty Fellowship, and gifts from Adobe, Amazon AWS, Google, and Snap. We would also like to thank Imperial Institute of Advanced Technology for GPU supports.

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
