[Supplementary Material]

# Supplemental Material for "Generative 3D Part Assembly via Dynamic Graph Learning"

This document provides the additional supplemental material that cannot be included into the main paper due to its page limit:

- Additional ablation study.
- Analysis of dynamic graph on additional parts and object categories.
- Training details.
- Failure cases and future work.
- Additional results of structural variation.
- Additional qualitative results.

## 1 Additional ablation study

In this section, we demonstrate the effectiveness of different components. We test our framework by proposing the following variants, where the results are in Table 1.

- **Our backbone w/o graph learning**: Replacing the graph learning module with a multi-layer perception to estimate the part-wise pose from the concatenated separate and overall part features.
- **Our backbone**: Only the iterative graph learning module.
- **Our backbone + relation reasoning**: Incorporate the dynamic relation reasoning module to our backbone.
- **Our backbone + part aggregation**: Incorporate the dynamic part aggregation module to our backbone.
- **Exchange dense/sparse node set iteration**: Switching the node set order in our algorithm. Specifically, we learn over the dense and sparse node set at even and odd steps respectively.
- **Input GT adjacency relation**: Instead of learning the relation weights from the output poses, we replace the dynamically-evolving relation weights with static ground truth adjacency relation between two parts, which is a binary value. Note the ground truth relation only covers the adjacency, not the symmetry and any other type of relations.
- **Reasoning relation from geometry**: We modify the dynamic relation reasoning module by replacing the input pose information in Equation 5 of the paper with part point cloud transformed by the estimated pose.

From Table 1, we observe that the proposed iterative GNN backbone, dynamic relation reasoning module and dynamic part aggregation module all contribute to the assembly quality significantly. Experimentally, we also exchange the order of sparse and dense node set in the iterative graph learning process, and do not observe much difference compared to our full algorithm. In order to justify our learned relation weights, we employ the ground truth binary adjacency relations to replace learned ones, and observe much worse performance than our learned relations. Finally, instead of learning the relation from estimated poses as in Equation 5 of the main paper, we alternatively replace the pose with transformed part point cloud, and also observe degraded performance as analyzed in the paper.

|                                        | Shape CD ↓ | PA ↑  | CA ↑  |
| -------------------------------------- | ---------- | ----- | ----- |
| Our backbone w/o graph learning        | 0.0086     | 26.05 | 28.07 |
| Our backbone                           | 0.0055     | 42.09 | 35.87 |
| Our backbone + relation reasoning      | 0.0052     | 46.85 | 38.60 |
| Our backbone + part aggregation        | 0.0051     | 48.01 | 38.13 |
| Exchange dense/sparse node set iteration | 0.0052   | 49.19 | 39.62 |
| Input GT adjacency relation            | 0.0053     | 45.43 | 35.66 |
| Reasoning relation from geometry       | 0.0053     | 45.11 | 39.21 |
| Our full algorithm                     | **0.0050** | **49.51** | **39.96** |

Table 1: Ablation study to demonstrate the effectiveness of each component of our algorithm. Shape CD, PA and CA are short for Shape Chamfer Distance, Part Accuracy and Connectivity Accuracy respectively.

Figure 1: Demonstration of the learned relation weights on additional parts and object categories. The number denotes the weight emitted from or received by the specific part averaged over all the other parts. The orange color is the top three or one relation weight in each column.

## 2 Additional analysis of the dynamic graph

We demonstrate additional learned relation weights in Figure 1. In the chair category, we expend the four parts in the main paper to eight parts, and we observe that the central parts (back, seat, head) have larger emitted relations and the peripheral parts (regular_leg, arm, footrest, pedestal_base, star_leg) have larger received relations. It reveals the same fact as shown in the main paper that the central parts guide the assembly process. Similar phenomenon can also be observed in the lamp and table categories, which demonstrate only four parts due to the limited common parts existing in the dataset.

## 3 Training details

Our framework is implemented with Pytorch. The network is trained for around 200 epochs to converge. The initial learning rate is set as 0.001, and we employ Adam to optimize the whole framework. We append the supervision on the output poses from all the time steps to accelerate the convergence. The graph neural network loops in five iterations for the final output pose. Experimentally, we find out that the performance tends to approach saturation for five iterations, while we haven't observed obvious improvement with more iterations.

In order to compute geometrically-equivalent parts, we firstly filter out the parts whose dimension difference of Axis-Aligned-Bounding-Boxes is above a threshold of 0.1, then cope with the remaining parts by excluding all the pairwise parts whose chamfer distance is below an empirical threshold of 0.2.

## 4 Failure cases and future work

In Figure 2, we show a few cases where our algorithm fails to assemble a well-connected shape and hence generates some floating parts. For example, the legs and arms are disconnected/misaligned from the chair base and back. This indicates the fact that our algorithm design lacks the physical constraints to enforce the connection among the parts. Our learned dynamic part graph builds a soft relation between the central and peripheral parts for a progressive part assembly procedure, but is short of the hard connection constraints. In the future work, we plan to solve this problem by developing a joint-centric assembly framework to focus more on the relative displacement and rotation between the parts, to facilitate the current part-centric algorithm.

Ground truth    Ours    Ground truth    Ours    Ground truth    Ours

Figure 2: Failure cases where some parts are floating and disconnected from the other parts.

## 5 Additional results of structural variation

Many previous works [1, 4, 3, 2] learn to create a novel shape from scratch by embedding both the geometric and structural diversity into the generative networks. However, provided the part geometry, our problem only allows structural diversity to be modeled. It poses a bigger challenge to the generative model since some shapes may be assembled differently, while the others can only be assembled uniquely (*i.e.,* only one deterministic result).

We demonstrate additional diverse assembled shapes with our algorithm in Figure 3. In the top five visual examples, it exhibits various results of structural variation, while in the bottom three examples, our algorithm learns to predict the same assembly results due to the limited input part set.

Assembly 1     Assembly 2     Assembly 3     Ground truth

Figure 3: Additional diverse results generated by our network. Top: structural variation demonstrated in part assembly; Bottom: no structural variation for the case with very limited parts.

# 6 Additional qualitative results

We demonstrate additional visual results predicted by our dynamic graph learning algorithm in Figure 4, 5 and 6.

B-Complement     B-Global     B-LSTM     Ours     Ground truth

Figure 4: Additional qualitative comparison between our algorithm and the baseline methods on the PartNet Chair dataset

B-Complement    B-Global    B-LSTM    Ours    Ground truth

Figure 5: Additional qualitative comparison between our algorithm and the baseline methods on the PartNet Table dataset

Figure 6: Additional qualitative comparison between our algorithm and the baseline methods on the PartNet Lamp dataset