[Reviews · NeurIPS 2020]

Review 1

Summary and Contributions: the paper proposes a network that assembles man-made shapes given the set of its parts. The network is computes SE(3) transformations for each part such that the unity of all parts generates the original shape. The main insight is that the parts of a shape can be represented as a graph, enabling the use graph neural networks, but that the connectivity of should correspond to the way the parts are positioned in the shape.

Strengths: - the paper presents a novel problem, and an interesting and creative solution. - being the first, the paper also proposes interesting a non-trivial alternatives to the problem as baselines for comparison, hence justifying the choice of the approach.

Weaknesses: - the paper could still use some more justification of design choices. How does the amount of iterations? - how well does the system to different label of details? Different number of points? - Some of the parts of the system are rather naive. For example, the use of a vanilla pointNet, or the use of q for r. Many other alternatives should be considered, but I am fine with this being in future work. After reading the rebuttal and other reviews, my opinions have not changed much. I think the experiments described in the rebuttal (both that answer my questions, and those of the others) are interesting to the reader, and should be added to the final revision.

Correctness: yes

Clarity: yes

Relation to Prior Work: Some of the tree-based networks are missing, which I find to be related. For example: "GAN-Tree: An Incrementally Learned Hierarchical Generative Framework for Multi-Modal Data Distributions"

Reproducibility: Yes

Additional Feedback:


Review 2

Summary and Contributions: This paper tackles the problem of generative 3D part assembly, which predicts a 6-doF part pose for each input part that assembles into a single 3D shape. The paper proposes a graph learning framework that uses an iterative graph neural network to adjust the relations between parts and infer their poses.

Strengths: + The task is meaningful but it is also relatively under-investigated in the 3D community. + The method is well motivated. Predicting the spatial relationships between object parts by graph inference is a natural idea. + The experimental results look promising, although some results need to be clarified/discussed more.

Weaknesses: My biggest concern is the part aggregation module: - I'm not sure if max-pooling is the best way to aggregate the node attributes among the parts into a single node. If we want to captuer the commonalities between different nodes, wouldn't average pooling be more appropriate? Some comparison is probably needed. - How should we interprete the relation between an aggregated part and the other parts? - In the ablation study, is "Our backbone w. relation reasoning" the model with aggregation module? - It would be great if the authors could discuss more about the results in Figure 3. First, the relations look inconsistent. For example, seat is strongly related to back while back is least related to seat; arm is most related to leg but leg is hardly related to arm. Second, why is leg most related to leg, given that they are not connected? Is it the effect of the aggregation module?

Correctness: The method looks correct, although I haven't checked the maths in detail.

Clarity: The paper is reasonably clear, but some notations could be improved. For example, in L168-169, using t and t+1 to represent odd and even numbers is informal. In Eq 6, instead of "k in V", it should be "v_k in V".

Relation to Prior Work: The paper explains the difference in details, but it could be further clarified. In L30-32, the authords mention that some previous work "assume certain part priors, such as a known number of parts", and "we assume no semantic knowledge upon the input parts". However, in L95 it says that "we assume to know the part count in each group."

Reproducibility: Yes

Additional Feedback: I read the rebuttal and comments from other reviewers, and I am leaning towards acceptance.


Review 3

Summary and Contributions: This paper aims to solve the 3D part assembly problem of a set of shape parts in a given 3D point cloud representation. The authors propose to apply an iterative graph neural network as a backbone, and predict part assembly in a coarse-to-fine way. Both the quantitative and qualitative results are promising.

Strengths: 1. The proposed dynamic graph learning framework is novel and technically sound. 2. Both the quantitative and qualitative results are convincing compared to the proposed three baselines. 3. The ablation study in Table 2 reflects the efficacy of using graph learning and relation reasoning. The additional ablation study provided in the supplemental is detailed.

Weaknesses: My main concern is the lack of clarification on the experimental details of the baseline method. It's not clear to me that whether the baseline methods are trained with the same losses, with the same termination strategy (same number of epochs, or stop training if achieving the best score on the validation set). A minor concern is the missing ablation study of the training losses. I'm surprised that without a connectivity loss/constraint, by directly optimizing the pose of each part, the assembly results are visually reasonable. It's better to have more discussions related to this. After reading the rebuttal, my concerns are mostly resolved.

Correctness: The proposed dynamic graph learning framework is technically sound.

Clarity: The paper is easy to follow.

Relation to Prior Work: Yes.

Reproducibility: No

Additional Feedback: Given my above concerns, I would rate the paper as marginally above the threshold. I will re-evaluate the scores based on the authors' feedback. Update: after reading the rebuttal, my concerns are mostly resolved. I'm inclined to accept.


Review 4

Summary and Contributions: The paper presents a solid work on conditioned shape generation, which is a quite massive subfield of 3D computer vision. In the same time authors show that their setting with parts assembly is different from related methods in that they use rigid part point clouds without any prior semantic knowledge.

Strengths: The method in the paper leverages a quite efficient and popular combination of dynamic graph and coarse-to-fine learning, which already showed good results in predicting relationships between objects in 3D. As there is no complete shape in input, the novelty of the work is clear.

Weaknesses: 1. It is hard to adapt existing works with 3D shape generation (like GRASS, PointEdit, SAGNet) to the current setting. 2. Maybe it could be also useful to see timings on network inference. 3. It is very interesting to see what would it be if one makes a slight modification and replaces the 6-DoF pose prediction with 9-DoF, that is augmented with scale vector prediction.

Correctness: Experimental section seems comprehensive. However, what first comes to mind is that proposed metrics are only focused on comparison with ground truth. However, we can assemble parts into plausible shapes in a large number of ways. In paper we can see some examples where baseline methods (and also the proposed) predict shapes that are different from ground truth, but at same time similar to real-world objects. No perceptual metrics are used paper to estimate the visual quality of shapes, which is as important as fitting accuracy.

Clarity: The paper is organized very well, the structure is clear.

Relation to Prior Work: There are several details in the problem setting that can identify it as a new problem formulation.

Reproducibility: Yes

Additional Feedback:

[Author Response · NeurIPS 2020]

We thank all the reviewers for their positive comments and constructive suggestions. We will incorporate the suggested changes and add the relevant references in the final revision, if accepted. Below, we address the primary questions.

**Response To Reviewer #1**

Q1: The number of iterations? We have conducted experiments of our full algorithm with 3 to 7 iterations, and observed worse performance after 5 iterations (Part accuracy: 45.51/49.51/44.6 for 3/5/7 iterations on the Table category), so we finally set that number to 5 in our paper. We find empirically that having too many iterations, makes the network harder to train and difficult to convergence. We will add a discussion in the final revision.

Q2: Different number of points per part? We have tried to sample 500 points per part. The performance remains roughly unchanged (PA: 48.78) with 500 points compared to the setting of sampling 1000 points (PA: 49.51) in our paper. This indicates that as long as the point samples are sufficient to cover the part geometric details, the performance does not change too much with increased sampling.

**Response To Reviewer #2**

Q1: Comparison to average pooling? We observed no significant performance difference between the average and max pooling (PA: 48.92 v.s. 49.51), indicating that both ways work. We will add the comparison in the revision.

Q2: Relation between an aggregated part and the other parts? We demonstrate this relation via the visualization in Fig.3 (the odd steps). For odd iterations, the relation graph is updated from the sparse node set, where the geometrically-equivalent parts, such as legs, are aggregated to a single node, and thus have relatively more influence as compared to the other parts. This is done the opposite way for the even iterations. We will make this clearer in the final paper.

Q3: Is"Our backbone w. relation reasoning" the model with aggregation module? No. It refers to the dynamic relation reasoning module mentioned in Sec 3.2. Further ablation study about the part aggregation is in the supplementary.

Q4: Discuss more about the results in Fig.3? The part relationship edges in our part graph are directional, so the relation from part A to part B can be different from the relation from B to A. So, in your examples, empirically we find that seat(arm) is more influential to back(leg), but it's not the case in the opposite direction. Legs are usually symmetric, so it is very likely that they share the same/similar part orientations, which explains their strong correlations.

Q5: Known part count or semantics? Previous works usually assume a known part count or semantics for the entire category of shapes (e.g., for chairs, four semantic parts: back, seat, leg and arm), while our algorithm does not rely on such an assumption. Our input is a set of parts, and the part count is computed purely from the geometry of the parts, so it can vary across different shapes. We will explain this more fully in the final paper.

**Response To Reviewer #3**

Q1: Experimental details for baselines, training strategies? The baseline methods are trained using the same losses and the same termination strategy as our method. We stop training when they achieve the best scores on the validation set. We will add more details in the final paper and release the code to the public.

Q2: Missing ablation of training losses? Experimentally, we find that removing any of the three losses (PA: w/o $\mathcal{L}_t/\mathcal{L}_r/\mathcal{L}_s$: 24.48/46.94/48.23) is less satisfactory compared to the full version (PA: 49.51).

Q3: Discussion on part connectivity? In Sec.4 of supplementary, we presented and discussed some failure examples with part dis-connection artifacts. Even though our method implicitly considers part relationships, we agree that more explicit handling of the part connectivity may further improve the results. We leave this as a future work.

**Response To Reviewer #4**

Q1: Adapt existing 3D shape generation works? Our task of part assembly takes as inputs a given set of parts to assemble and outputs the part poses for each part. This is quite different from the 3D shape generation tasks, where the part geometry is generated as outputs. However, there are some shared techniques, e.g., handling of part relationships.

Q2: Timings on network inference? The network inference roughly takes 0.3 second per example, which is very fast. In the revision, we can add more thorough comparisons on timings if requested.

Q3: Replace the 6-DoF pose prediction with 9-DoF? Our problem is motivated by the practical scenario of assembling an IKEA furniture, where the scale for each part is fixed by definition. However, we agree with the reviewer that it could be interesting to apply our method for other analogous problems that require 9-DoF prediction.

Q4: Only compare with the ground truth? As indicated in Line 226-228, when we evaluated the performance of these methods, we generated multiple possible assembly outputs with different Gaussian random noise, and measured the minimum distance from the assembly predictions to the ground truth, which allows the models to generate shapes that are different from ground truth while similar to real-world objects. We can make this clearer in the final paper.

[Meta-Review · NeurIPS 2020]

All four reviewers were positively inclined about this paper, with the primary weaknesses being fairly low-level and minor concerns. The AC is inclined to agree with the reviewers. Given that the rebuttal addressed a number of concerns, the AC encourages the authors to incorporate information from their rebuttal in the camera-ready.